# Processes for creating an interprofessional mental health identity among pre-registration healthcare students: A scoping review protocol

Melissa Owens[1], Gérard Charl Filies[2], Molly Crosland[1], Claire Carswell[1]*

1 Department of Health Sciences, University of York, York, United Kingdom, 2 Interprofessional Education Unit, Faculty of Community and Health Sciences, University of the Western Cape, Cape Town, Western Cape, South Africa

* claire.carswell@york.ac.uk

## Abstract

### Background

Developing an interprofessional mental health identity among pre-registration (licensure) healthcare students is critical for promoting effective interprofessional collaboration in mental health care. An interprofessional mental health identity refers to the shared beliefs, values and attitudes that will enable healthcare students/professionals to work together effectively in addressing mental health issues and foster interprofessional collaborative person-centred practice. However, the processes and strategies for facilitating this amongst pre-registration healthcare students are not clearly defined or understood. This scoping review aims to map the available literature on how interprofessional mental health identity is created for pre-registration healthcare students.

### Methods

We will follow the five stages of Arksey and O'Malley's Framework and identify our search terms using the 'population', 'concept' and 'context' (PCC) criteria. We will search CINAHL, Medline, British Nursing Database, PsycInfo, Cochrane Library and AMED from 1990 to May 2024. Retrieved records will be managed in Covidence and screened independently by two reviewers. Data extraction forms will be developed to capture relevant data related to the review aim. The forms will be piloted, and two independent reviewers will complete the data extraction process. Narrative synthesis will be used to provide a descriptive overview of the included articles.

### Discussion

Little is known about the process by which pre-registration healthcare students develop an interprofessional mental health identity. Therefore, this scoping review

**Data availability statement:** No datasets were generated or analysed during the current study. All relevant data from this study will be made available upon study completion.

**Funding:** The author(s) received no specific funding for this work.

**Competing interests:** The authors have declared that no competing interests exist.

will draw together existing research to create a conceptual model for the process of developing an interprofessional mental health identity.

## Introduction

Mental health is a neglected but crucial aspect of overall well-being, but its importance in healthcare has been increasingly recognised in recent years. Votruba and Thornicroft (2016) [1] highlight the growing importance of mental health in healthcare. In September 2015, mental health was included in the United Nations (UN) Sustainable Development Goals (SDGs). The UN, therefore, acknowledged that for the next 15 years, the burden of disease of mental illness is a growing priority for global development. Nevertheless, mental health orders remain at an all-time high, and access to effective mental health services continues to be a significant challenge globally for the population [2]. Furthermore, it is recognised that many healthcare systems across the world continue to provide fragmented care, but that healthcare professionals need to be able to work together collaboratively, if this is to be addressed.

In 2010, the World Health Organisation (WHO) launched its framework for action on interprofessional education (IPE) and collaborative practice (CP), emphasising the importance of pre-registration health care professionals being 'collaborative practice ready' at the point of registration [2]. Its importance is recognised in enhancing communication across professions, whilst increasing the likelihood of shared decision-making, providing a safe environment for advice seeking and admitting mistakes. The definition of CP, however, lacks universal clarity [3] but is often described as both a system and person-level approach to working together to address the complexity of fragmented care [4] to enhance the quality and safety of the care provided [5–9].

IPE is a global, pedagogical approach considered to be best practice for equipping students with the knowledge, skills, behaviours and attitudes for CP and to develop a shared understanding of their roles and responsibilities in providing patient-centred care [2]. It is defined as: 'occasions when members or students of two or more professions learn with, from and about each other to improve collaboration and the quality of care' [10]. For this reason, IPE and CP are inextricably linked and frequently referred to as IPECP. In relation to mental health, it is identified as a key action for improving the quality and efficiency of mental health services [4].

However, just as there is no globally shared agreement as to the meaning of CP, there is also no shared agreement as to the precise knowledge, skills, attitudes and behaviours that students should develop through IPE, to effectively equip them to become 'collaborative practice ready' [2]. The United States and Canada, for example, both have their own sets of competencies they identify as necessary in individual students [4]. Australia published a capability framework in 2011, which highlighted three core elements: client-centred service, client safety, and collaborative practice. These core elements are underpinned by five collaborative practice capabilities [11]. Others, such as the United Kingdom, are attempting to create a set of standards for

the delivery of effective IPE that align with competencies identified by different regulatory bodies [10]. Nonetheless, whilst there is no global agreement on its precise definition, there remains universal recognition of its importance.

According to Reinders et al. [9], identity can be best described as the sense of oneself, shaped by one's own personality, experiences, and social interactions. Social identity, however, refers to the association of oneself with a specific group. Extended identity theory builds on these theories and forms the basis for the concept of 'interprofessional identity', arguing that an interprofessional identity needs to be built alongside the professional identity, without impacting on it [9].

Whilst no definitive definition of 'interprofessional identity' exists, it is helpfully described by Khalili et al. [12] as a 'sense of belonging to your own profession and interprofessional community'. It is made up of three elements: a sense of belonging, commitment and shared beliefs [13]. It is also described as a set of shared beliefs, values, and attitudes that enable healthcare students or professionals to work together effectively, making it an essential part of CP [12]. Discussions are continuing regarding how best to facilitate the development of an interprofessional identity, with challenges such as power dynamics as a factor that can impede it [6].

However, pre-registration students face particular challenges in finding a mental health field identity due to the generic nature of most professional programmes. Globally, for example, nursing students will complete a generic programme of study and then specialise in a field of practice after registration. In South Africa, it is common practice for pre-registration students to be placed at facilities that focus on mental health challenges, to allow students to gain practical experience as part of their workplace learning. However, in the UK, nursing students choose their field of practice at the point of application, meaning the professional identity for mental health field students could be formed early in their professional careers.

Yet, a significant challenge they face is the genericism of the nursing programmes within the UK. The Professional Standards for Nursing [14] require students from all fields of nursing to be competent in the same proficiencies, with some arguing that there is great emphasis and value placed on the acquisition of physical skills over those more closely aligned to mental health, which can lead to a sense of hierarchy between fields of nursing, with mental health nursing seen as being of lower status than that of the adult field nurse [15].

It is generally recognised that pre-registration students have a strong desire to 'belong' to their own profession before and in the early stages of their professional journey [5,16]. However, this can lead to resentment of others [17]. And could, therefore, be of particular concern for those in the field of mental health. Nevertheless, having a strong professional identity is considered an essential prerequisite to developing a strong interprofessional identity [9,12]. However, it can conversely create a barrier to it if the uni-professional identity feels threatened [17]. It is, therefore, important to understand and identify the processes that have been used to develop an interprofessional identity before evaluating the best or most effective approach in this context.

To establish a foundational understanding of the literature on the subject, it is essential first to identify the literature that has been published in this field and map the available evidence to identify any key gaps in our knowledge. Therefore, a scoping review, which enables us to explore the breadth of the available evidence, is a necessary first step. Although there appears to be a lack of literature identifying, specifically, what an interprofessional mental health identity would look like, the particular challenges faced by students working in this field make this review timely.

## Aim and objectives

This review aims to map the available literature on how interprofessional mental health identity has been created for pre-registration healthcare students and includes two overarching objectives:

• Outline the range and nature of literature published on interprofessional mental health identity for pre-registration healthcare students.

• Describe the processes and strategies used to foster an interprofessional mental health identity in pre-registration healthcare students.

## Materials and methods

A scoping review will be conducted to address the breadth of the research question and map the available literature on the subject. This scoping review will follow the five stages of Arksey and O'Malley's (2005) Framework [18] and be reported in line with the PRISMA-ScR reporting guidelines [19]. The protocol has been registered prospectively on Open Science Framework Registries (DOI: https://doi.org/10.17605/OSF.IO/TJ9E7) prior to screening records. At the time of writing this protocol, data extraction has not yet started.

### Stage one: Identification of the research question

The research question that this review aims to answer is 'What is known from the existing literature about how an interprofessional mental health identity is developed among pre-registration healthcare students, including the range, nature, and strategies used to develop this identity?'.

### Stage two: Identify relevant studies

**Eligibility criteria.** Inclusion and exclusion criteria were developed according to the PCC mnemonic (Population, Concept, Context).

We will include:

- Articles that focus on a population of pre-registration healthcare students or involve participants who are pre-registration healthcare students. In this review, this is defined as anyone undertaking a degree in higher education that would make them eligible for registration with a professional regulatory body, enabling them to work in the area of mental health. This would include, but is not limited to, nurses, medical doctors, occupational therapists, pharmacists, other allied healthcare professionals, clinical psychologists and social workers.

- Articles which describe interprofessional mental health identity among pre-registration healthcare students.

- Articles which describe the processes of creating an interprofessional mental health identity.

- Primary empirical research articles (using any methodology), opinion pieces, theoretical articles, news articles, governmental or charity reports, white papers, editorial letters, commentary, and other grey literature.

- Articles published in English.

- Articles published after 1990.

We will exclude:

- Articles which focus exclusively on post-registration healthcare professionals.

- Articles that focus exclusively on healthcare workers or students enrolled in a course or degree that does not make them eligible for registration with a professional regulatory body, or who will not be subjected to statutory regulation before May 2024. For example, physician associates, or healthcare support workers.

- Articles which describe the processes of creating an interprofessional identity outside the context of mental health.

- Articles published in a language other than English, due to a lack of resources for translation services.

- Articles published before 1990, due to the changes in education and regulation of healthcare workers prior to 1990.

**Information sources.** We will search the following electronic databases: CINAHL, Medline, British Nursing Database, PsycINFO, Cochrane Library, and AMED from 1990 to May 2024 using search terms relevant to the population, concept and context of the review. These search terms were developed by identifying the relevant population, concept, context,

and key terms that would capture these aspects of the review question. We also identified previous reviews in relevant areas to identify important terms and MeSH headings for inclusion in the search strings. We refined the search strings through consultation with a subject librarian and piloting the search strings on Medline and CINAHL. An exemplar search strategy for Medline is available in Appendix 1. We will identify grey literature through searches of EThOS and ProQuest to find relevant dissertations and theses, and search for evaluation, policy documents and white papers from relevant organisations, such as Interprofessional.Global (IP.Global), Interprofessional Research.Global (IPR.Global), The Centre for the Advancement for Interprofessional Education (CAIPE), The Africa Interprofessional Education Network (AfrIPEN), The Asia Pacific Interprofessional Education and Collaboration Network (APIPEC), the Arab Network for Interprofessional Collaboration (ANIC), The Australian Interprofessional Practice and Education Network (AIPPEN), The Canadian Interprofessional Health Collaborative (CHIC), The National Centre for Interprofessional Practice and Education (AIHC), The National Centre for Interprofessional Practice and Education (NEXUS) The Nordic Interprofessional Network (NIPNET), The Regional Network for Interprofessional Education in the Americas (REIP), The National Academies of Practice (NAP), The Dutch-speaking Network for Interprofessional Collaboration (IPINN), The Indian Interprofessional Education Network (IndIPEN), Interprofessional Education Collaborative (IPEC) and the Society for interprofessionalism in Healthcare (IP-Health).

We will hand-search specialist journals that publish research on interprofessional working in health and social care, such as the Journal of Interprofessional Care, and consult with relevant experts in the field to ensure that all relevant articles are included in the review. Finally, we will review the reference list of included articles to identify any other publications that have not been identified through previous searches.

## Stage 3: Identify relevant studies

Identified records will be imported into Covidence. Following the removal of duplicates, two reviewers will independently screen titles and abstracts according to the prespecified eligibility criteria. Any disagreements will be resolved through discussion or referral to a third reviewer.

Following title and abstract screening, full texts will be sourced where possible for the potentially eligible records. Two independent reviewers will then read and screen the full texts against the eligibility criteria. Any disagreements will be resolved through discussion or referral to a third reviewer.

The whole screening process will be presented in a PRISMA flow chart. Studies with multiple publications will be grouped and reported as single studies under a primary reference.

## Stage 4: Charting the data

**Data charting process.** Data charting forms will be developed to capture all relevant data items related to the review's aim. Two reviewers will pilot test these forms and conduct the data extraction process.

**Data items.** To address the review aims, the following data items will be extracted from included articles: author(s), year of publication, country of publication or data collection, aims/purpose, population and sample size within the source of evidence (if applicable), methodology, key findings that relate to the scoping review question/s, curricula activities, interventions, disciplines, and process/strategy used for IP MH identity formation.

## Stage 5: Collating, summarising and reporting the results

**Synthesis of results.** We will carry out a narrative synthesis to present a descriptive overview of the included articles. This synthesis will identify key themes related to the strategies used to develop an interprofessional mental health identity, including an evaluation of these strategies in the available literature. These themes may include barriers, facilitators and specific curricula activities used in the process. We will then group these themes from the synthesis into broader

conceptual categories and map the relationships between these concepts to develop a conceptual model for the process of creating an interprofessional field identity in mental health.

## Discussion

Through reviewing the literature, this scoping review will highlight those aspects that are essential to help achieve an interprofessional mental health identity in pre-registration students. To date, research on interprofessional identity has focused on an identity across professions, rather than an identity across the mental health field, and particularly at the pre-registration level. Through undertaking this review, it is anticipated that a theoretical model to help facilitate an interprofessional mental health identity can be proposed.

## Supporting information

**S1 Appendix.  Medline search strategy.**
(DOCX)

**S1 File.  PRISMA-P Checklist for protocols.**
(DOCX)

## Author contributions

**Conceptualization:** Melissa Owens.

**Data curation:** Melissa Owens, Claire Carswell.

**Investigation:** Molly Crosland.

**Methodology:** Melissa Owens, Gerard Charl Filies, Molly Crosland, Claire Carswell.

**Project administration:** Melissa Owens, Claire Carswell.

**Resources:** Melissa Owens.

**Software:** Claire Carswell.

**Supervision:** Melissa Owens.

**Writing – original draft:** Melissa Owens, Gerard Charl Filies, Claire Carswell.

**Writing – review & editing:** Melissa Owens, Gerard Charl Filies, Molly Crosland, Claire Carswell.

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
