## [Decision Letter · Decision Letter 0]

26 Jun 2025

Dear Dr. Carswell,

Thank you for submitting your manuscript to PLOS ONE. After careful consideration, we feel that it has merit but does not fully meet PLOS ONE’s publication criteria as it currently stands. Therefore, we invite you to submit a revised version of the manuscript that addresses the points raised during the review process.

**The conflicting reviews from the 2 reviewers suggest that the authors should respond to each point separately. This will help clear up any confusion and address all concerns. However, if the authors do not agree with the reviewers' suggestions, they should explain why, using clear reasons and evidence. Thank you.**

We look forward to receiving your revised manuscript.

Kind regards,

Filipe Prazeres, MD, MSc, Ph.D.

Academic Editor

PLOS ONE

**Journal Requirements:**

1. When submitting your revision, we need you to address these additional requirements. Please ensure that your manuscript meets PLOS ONE's style requirements, including those for file naming. The PLOS ONE style templates can be found at https://journals.plos.org/plosone/s/file?id=wjVg/PLOSOne_formatting_sample_main_body.pdf and https://journals.plos.org/plosone/s/file?id=ba62/PLOSOne_formatting_sample_title_authors_affiliations.pdf 2. Please include captions for your Supporting Information files at the end of your manuscript, and update any in-text citations to match accordingly. Please see our Supporting Information guidelines for more information: http://journals.plos.org/plosone/s/supporting-information.

Reviewers' comments:

Reviewer's Responses to Questions

**Comments to the Author**

1. Does the manuscript provide a valid rationale for the proposed study, with clearly identified and justified research questions?

Reviewer #1: Yes

Reviewer #2: Partly

2. Is the protocol technically sound and planned in a manner that will lead to a meaningful outcome and allow testing the stated hypotheses?

Reviewer #1: Yes

Reviewer #2: Partly

3. Is the methodology feasible and described in sufficient detail to allow the work to be replicable?

Reviewer #1: Yes

Reviewer #2: No

4. Have the authors described where all data underlying the findings will be made available when the study is complete?

Reviewer #1: Yes

Reviewer #2: Yes

5. Is the manuscript presented in an intelligible fashion and written in standard English?

Reviewer #1: Yes

Reviewer #2: No

You may also provide optional suggestions and comments to authors that they might find helpful in planning their study.

**Reviewer #1: ** Review Report for PONE-D-25-20588

Manuscript Title: Processes for creating an interprofessional mental health identity among pre-registration healthcare students: A scoping review protocol.

Registration: The protocol has been prospectively registered on Open Science Framework Registries (DOI: https://doi.org/10.17605/OSF.IO/TJ9E7).

Overall Recommendation: Revision.

Rationale for Recommendation:

The submission of a scoping review protocol is appropriate for publication in PLOS ONE, as the journal explicitly publishes "Study Protocols" that detail plans for research projects yet to be conducted. This aligns with PLOS ONE's emphasis on scientific validity, robust methodology, and ethical standards. Publishing a protocol enhances research transparency, reduces potential bias, and acknowledges the foundational work before data collection.

However, this protocol requires revision to ensure it fully meets PLOS ONE's expectations for methodological rigor, clarity, and to adequately differentiate its contribution from other review types.

The "Background" section touches upon the concept of interprofessional identity and the gap in understanding how it's formed in the mental health context for pre-registration students. This provides a good foundation for the necessity of a scoping review.

The "Discussion" section briefly reiterates the knowledge gap and the expected outcome of a conceptual model.

Recommended Revisions:

Strengthen the Introduction/Background: While the background establishes the need, it would be beneficial to explicitly and concisely define scoping reviews and their purpose, clearly differentiating them from systematic reviews. This will further justify the chosen methodology and highlight its specific contribution.

Elaborate on Data Synthesis: Although narrative synthesis is mentioned, provide more detail on how the charted data will be analyzed and synthesized to achieve the stated objectives, particularly how it will lead to the development of a conceptual model. Clarify the analytical approach to moving from charted data to a proposed model.

Explicitly Address Reporting Guidelines: While the protocol mentions a PRISMA flow chart for the screening process, it would be beneficial to mention that the final scoping review will adhere to relevant reporting guidelines, such as PRISMA-ScR (Preferred Reporting Items for Systematic Reviews and Meta-Analyses extension for Scoping Reviews), to ensure comprehensive and transparent reporting of the eventual findings.

Consider Ethical Statement Nuance: While "N/A" for an ethics statement is permissible, briefly stating why it's not applicable (e.g., "This protocol outlines a review of existing literature and does not involve direct human participants or identifiable data, thus formal ethical approval is not required") could add clarity and preempt potential questions.

By addressing these suggested revisions, the authors can significantly enhance the clarity, rigor, and overall impact of their protocol, demonstrating its valuable contribution to the field in line with PLOS ONE's publication standards.

**Reviewer #2:**  The value of this study speaks for itself, unfortunately it is not fully described in this protocol. I encourage authors to complete the scoping review and work to clarify the issues described below.

I recommend establishing what interprofessional collaboration means in this context somewhere in the first paragraph. Perhaps line 57.

An example of the type of interprofessional mental health identity would be helpful. What types of beliefs, values, and attitudes have been cited to be helpful?

Lines 70-73 The description of interprofessional identity is unclear to me. I think these lines could use clarification. The ideas spanning the two sentences on lines 73 are not connected. Same comment on line 75. More information here would be helpful. Are power dynamics the only barrier to creating more harmonious interprofessional working practices?

Paragraphs starting at lines 83 and 90 should be combined. The ideas therein are too similar.

The introduction is not a sufficient review of the literature nor is it publication ready. Many ideas are disconnected and not organized in a clear logical manner.

The research question in the methods section does not align with the objectives stated prior.

Line 194 Charting should be capitalized

Line 206 Collating should be capitalized

Narrative synthesis needs to be described fully. This is not enough information.

Line 213 typos

Many typos throughout manuscript.

**Do you want your identity to be public for this peer review?** For information about this choice, including consent withdrawal, please see our Privacy Policy

Reviewer #1: **Yes: ** Hailemariam Mamo Hassen(PhD)

Reviewer #2: No

---

## [Author Response · Author response to Decision Letter 1]

11 Aug 2025

Reviewer 1

The submission of a scoping review protocol is appropriate for publication in PLOS ONE, as the journal explicitly publishes "Study Protocols" that detail plans for research projects yet to be conducted. This aligns with PLOS ONE's emphasis on scientific validity, robust methodology, and ethical standards. Publishing a protocol enhances research transparency, reduces potential bias, and acknowledges the foundational work before data collection. However, this protocol requires revision to ensure it fully meets PLOS ONE's expectations for methodological rigor, clarity, and to adequately differentiate its contribution from other review types.

The "Background" section touches upon the concept of interprofessional identity and the gap in understanding how it's formed in the mental health context for pre-registration students. This provides a good foundation for the necessity of a scoping review.

The "Discussion" section briefly reiterates the knowledge gap and the expected outcome of a conceptual model

Strengthen the Introduction/Background: While the background establishes the need, it would be beneficial to explicitly and concisely define scoping reviews and their purpose, clearly differentiating them from systematic reviews. This will further justify the chosen methodology and highlight its specific contribution.

Thank you for your feedback, we have amended the manuscript to ensure it meets the requirements of the journal. We have included a justification for choosing a scoping review methodology at the end of the methodology section.

-

Elaborate on Data Synthesis: Although narrative synthesis is mentioned, provide more detail on how the charted data will be analyzed and synthesized to achieve the stated objectives, particularly how it will lead to the development of a conceptual model. Clarify the analytical approach to moving from charted data to a proposed model.

We have provided additional clarity on the data synthesis; however, as it is a scoping review, the analysis will be purely descriptive, as the aim is not to create generalizable findings.

-

Explicitly Address Reporting Guidelines: While the protocol mentions a PRISMA flow chart for the screening process, it would be beneficial to mention that the final scoping review will adhere to relevant reporting guidelines, such as PRISMA-ScR (Preferred Reporting Items for Systematic Reviews and Meta-Analyses extension for Scoping Reviews), to ensure comprehensive and transparent reporting of the eventual findings.

We have clarified that we will report the findings in line with PRISMA-ScR reporting guidelines.

-

Consider Ethical Statement Nuance: While "N/A" for an ethics statement is permissible, briefly stating why it's not applicable (e.g., "This protocol outlines a review of existing literature and does not involve direct human participants or identifiable data, thus formal ethical approval is not required") could add clarity and pre empt potential questions.

This response is in the meta data and as a scoping review an ethical statement is not applicable.

The application system specifies that if it is not applicable then ‘N/A’ should be entered.

Reviewer 2

The value of this study speaks for itself, unfortunately it is not fully described in this protocol. I encourage authors to complete the scoping review and work to clarify the issues described below.I recommend establishing what interprofessional collaboration means in this context somewhere in the first paragraph. Perhaps line 57.

Thank you for your comments

This has now been discussed from a global context, including the relationship between interprofessional education, collaborative practice and, interprofessional identity. In addressing the comments below, the sequencing has been altered.

-

An example of the type of interprofessional mental health identity would be helpful. What types of beliefs, values, and attitudes have been cited to be helpful?

There is a lack of literature identifying this and will hopefully be drawn out through the scoping review. However, additional information has been included as to why this is the case and why it is especially important for mental health.

-

Lines 70-73 The description of interprofessional identity is unclear to me. I think these lines could use clarification. The ideas spanning the two sentences on lines 73 are not connected. Same comment on line 75. More information here would be helpful. Are power dynamics the only barrier to creating more harmonious interprofessional working practices?

The sequencing of the arguments given have been adjusted. Additional examples of the benefits of IPC have been included.

-

Paragraphs starting at lines 83 and 90 should be combined. The ideas therein are too similar. The sequencing of points, overall, have been reviewed and realigned, including these. The introduction is not a sufficient review of the literature nor is it publication ready. Many ideas are disconnected and not organized in a clear logical manner.

The sequencing of the arguments given have been adjusted to help with the synergy of points made. A wider scope of literature has taken place and been included in the introduction.

-

The research question in the methods section does not align with the objectives stated prior.

This has been amended to more accurately reflect the aims and objectives of the review.

-

Line 194 Charting should be capitalized

This has been amended

-

Line 206 Collating should be capitalized

This has been amended.

-

Narrative synthesis needs to be described fully. This is not enough information.

We have added more detail to the section on synthesis.

-

Line 213 typos

Many typos throughout manuscript.

We have checked for grammar and typos using several spellchecks and hope that these have been identified and amended.

---

## [Decision Letter · Decision Letter 1]

16 Sep 2025

Dear Dr. Carswell,

Thank you for submitting your manuscript to PLOS ONE. After careful consideration, we feel that it has merit but does not fully meet PLOS ONE’s publication criteria as it currently stands. Therefore, we invite you to submit a revised version of the manuscript that addresses the points raised during the review process.

We look forward to receiving your revised manuscript.

Kind regards,

Filipe Prazeres, MD, MSc, Ph.D.

Academic Editor

PLOS ONE

Journal Requirements:

Reviewers' comments:

Reviewer's Responses to Questions

**Comments to the Author**

1. Does the manuscript provide a valid rationale for the proposed study, with clearly identified and justified research questions?

Reviewer #2: Yes

2. Is the protocol technically sound and planned in a manner that will lead to a meaningful outcome and allow testing the stated hypotheses?

Reviewer #2: Yes

3. Is the methodology feasible and described in sufficient detail to allow the work to be replicable?

Reviewer #2: Yes

4. Have the authors described where all data underlying the findings will be made available when the study is complete?

Reviewer #2: No

5. Is the manuscript presented in an intelligible fashion and written in standard English?

Reviewer #2: No

You may also provide optional suggestions and comments to authors that they might find helpful in planning their study.

Reviewer #2: Thank you for your revisions, they provided much needed clarity.

Unfortunately, there are still many typos throughout the protocol. I suggest having another person read through and flag additional spelling and grammar issues since the journal does not provide copyediting.

**Do you want your identity to be public for this peer review?** For information about this choice, including consent withdrawal, please see our Privacy Policy

Reviewer #2: No

---

## [Author Response · Author response to Decision Letter 2]

23 Sep 2025

Unfortunately, there are still many typos throughout the protocol. I suggest having another person read through and flag additional spelling and grammar issues since the journal does not provide copyediting

Thank you for highlighting this. The manuscript has been proof read and also put through Grammarly spell checker to ensure any typos have been flagged and corrected.

---

## [Editor Report · Decision Letter 2]

30 Sep 2025

Processes for creating an interprofessional mental health identity among pre-registration healthcare students: A scoping review protocol.

PONE-D-25-20588R2

Dear Dr. Carswell,

We’re pleased to inform you that your manuscript has been judged scientifically suitable for publication and will be formally accepted for publication once it meets all outstanding technical requirements.

Kind regards,

Filipe Prazeres, MD, MSc, Ph.D.

Academic Editor

PLOS ONE
---

## [Editor Report · Acceptance letter]

PONE-D-25-20588R2

PLOS ONE

Dear Dr. Carswell,

I'm pleased to inform you that your manuscript has been deemed suitable for publication in PLOS ONE. Congratulations! Your manuscript is now being handed over to our production team.

Kind regards,

on behalf of

Prof. Filipe Prazeres

Academic Editor

PLOS ONE